# Transforming Land Administration Practices through the Application of Fit-For-Purpose Technologies: Country Case Studies in Africa

**Danilo Antonio, Solomon Njogu \*, Hellen Nyamweru** 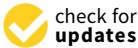 **and John Gitau** 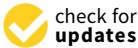

UN-Habitat/Global Land Tool Network, P.O. Box 30030, Nairobi 00100, Kenya; danilo.antonio@un.org (D.A.); hellen-nyamweru.ndungu@un.org (H.N.); john.gitau@un.org (J.G.)
\* Correspondence: solomon.njogu1@un.org

**Abstract:** Access to land for many people in Africa is insecure and continues to pose risks to poverty, hunger, forced evictions, and social conflicts. The delivery of land tenure in many cases has not been adequately addressed. Fit-for-purpose spatial frameworks need to be adapted to the context of a country based on simple, affordable, and incremental solutions toward addressing these challenges. This paper looked at three case studies on the use of the Social Tenure Domain Model (STDM) tool in promoting the development of a fit-for-purpose land administration spatial framework. Data gathering from primary and secondary sources was used to investigate the case studies. The empirical findings indicated that the use and application of the STDM in support of the fit-for-purpose land administration framework is quite effective and can facilitate the improvement in land tenure security. The findings also revealed that the tool, together with participatory and inclusive processes, has the potential to contribute to other frameworks of Fit-For-Purpose Land Administration (FFP LA) toward influencing changes in policy and institutional practices. Evidently, there was a remarkable improvement in the institutional arrangements and collaboration among different institutions, as well as a notable reduction in land conflicts or disputes in all three case studies.

**Keywords:** fit-for-purpose land administration; spatial framework; tenure security; STDM; technology

## 1. Introduction

Access to land for the majority of people, particularly in developing countries, is insecure and continues to pose risks to poverty, hunger, forced evictions, and social conflicts. The delivery of land tenure in many cases has not been adequately addressed, due to weak legal systems, complex procedures, and poor institutional arrangements [1,2]. Notably, in Sub-Saharan Africa, customary and informal land tenure systems dominate statutory forms of tenure and are generally not recorded in the official registry [3]. The growing competition for land is threatening customary land due to the population increase and pressure from large-scale land-based investment projects [4]. The risk of evictions and land grabbing is even higher with the increasing speculation of land that threatens the access and use of land among the poor and vulnerable groups, particularly women. Similarly, due to the high rate of urbanization, much of the growth is taking place within informal settlements linked to the challenge of affordability [5]. Most land administration systems have set aside the registration of informal, customary, and complex land rights due to ambiguities in the legal, socio-cultural, and technical standards, among others. Hence, access to land administration services for the poor is thus limited by the cost, capacity, complexity of procedures, and rigid technical requirements, depriving them the opportunity to access land that would otherwise enable them to invest toward improving incomes and livelihoods [6].

The benefit of well-defined, documented, and enforceable land rights cannot be underestimated. With the global commitment enshrined in the Sustainable Development Goals (SDGs), 'tenure security for all' is emphasized, especially for the poor and marginalized

groups, including women. The United Nations General Assembly through New Urban Agenda (NUA) commitment number 35 promotes secure tenure for all and the recognition of the continuum of land rights approach through fit-for-purpose solutions with a particular emphasis on women [7]. Similarly, Agenda 2063 of the African Union (AU Goal 17) aims to give 20% of rural women the access to and control of land by 2023 [8]. The African Union, while resolving the commitment to poverty eradication, agreed to ensure equitable access to land and related resources among all land users, including the youth and other landless and vulnerable groups such as displaced persons [9]. The Voluntary Guidelines on the Responsible Governance of Tenure of Land, Fisheries and Forests in the Context of National Food Security (VGGT) also recommend the application of pro-poor and gender-responsive land policies to achieve equitable development [10].

### 1.1. Innovative Approaches for Addressing Tenure Insecurity

Currently, it is estimated that less than 30% of land parcels are formally registered [11]. To address the land registration gap, locally suited methodologies based on the continuum of land rights approach are being promoted as alternative, effective, and scalable solutions in favor of conventional practices in land administration [12]. The principles of the fit-for-purpose land administration (FFP LA) approach complement these methodologies as building blocks in the way we record and perceive tenure security [13]. The FFP LA approach recognizes the need to be flexible in the spatial, institutional, and legal frameworks without being limited by stringent standards of technical accuracy and legal requirements [14]. Their application leverages sustainable practices in land administration that are the foundation for good land governance [15].

Evidently, it has been accepted that relying on a single form of tenure to meet the demands of different social groups will be cumbersome [16]. This follows the argument of Payne regarding the range of actions, steps, and decisions on land that can be interpreted as having meaning to land tenure security [17]. Essentially, the analogy of tenure security can be viewed in many forms even in social claims, and the assumption on this study alludes to those interpretations along the continuum of land rights approach. The Continuum of Land Rights advances this further by recognizing that land rights are complex or overlap and can be assumed to be lying on a continuum often able to transform at different times [18]. The Continuum of Land Rights advocates for the incremental approach in recording and recognizing diverse forms of tenure that may be statutory or social claims at the time of recording. The new Urban Agenda commitments 35 promotes this approach in delivering security of tenure for all, recognizing the plurality of tenure types, and developing fit-for-purpose solutions to suit the context.

The FFP LA principles reinforce the application of the Continuum of Land Rights approach through the use of appropriate technology such as the STDM [19]. The FFP LA spatial framework requires that land recording is aligned to the local requirements based on a simple, affordable, and incremental approach [20]. The International Federation of Surveyors (FIG) considers this approach as 'fit for purpose,' as it is based on locally available resources and capacities [13]. The use of technology coupled with freely available geospatial data enables the creation of a suitable infrastructure for the coordinated production, access, and use of geospatial data among producers and users in an electronic environment [21].

In this paper, the focus is on the use of the STDM tool in promoting the application of FFP LA and the Continuum of Land Rights approach, particularly in the development of the FFP LA spatial framework. The STDM tool has evolved from conceptualization to implementation in the last 15 years. To date, the STDM tool has been implemented (and is being implemented) in at least 15 countries in various contexts such as informal settlements [22,23], customary [24], post-conflict, and formal land registration [25]. Various studies in the literature have pointed out the need for the STDM tool, particularly toward supporting the continuum of land rights approach and in improving the land tenure security of poor people [26,27]. This study sought to demonstrate, through empirical findings, that the STDM tool, with the participatory and inclusive approaches, facilitates

tenure security improvement and other development outcomes. The paper also showed how the STDM tool, which is supporting the development of a spatial framework, is able to influence the incremental but transformative change in the policy and institutional aspects of FFP LA.

*1.2. The Social Tenure Domain Model*

The STDM is a concept and model that has been developed into an information tool to support FFP LA approaches and to overcome the barriers of conventional land administration systems [28,29]. The STDM model conforms to the global standards of the Land Administration Domain Model (LADM) and is meant to provide a flexible customization framework to suit local applications in different contexts [30]. Thus, the conceptual design of the STDM is robust and flexible to support FFP LA in the recording of both formal and informal tenure rights. The motivation has been to break the norm from the core cadastral thinking of parcels toward a range of spatial units and a broad view of tenure types, as perceived in the continuum of land rights. The development of STDM as a tool has been tested and accepted internationally for being practical, fast, and affordable in facilitating land tenure security especially for the poor and marginalized groups [31]. The conceptual model of the STDM is shown in Figure 1 below.

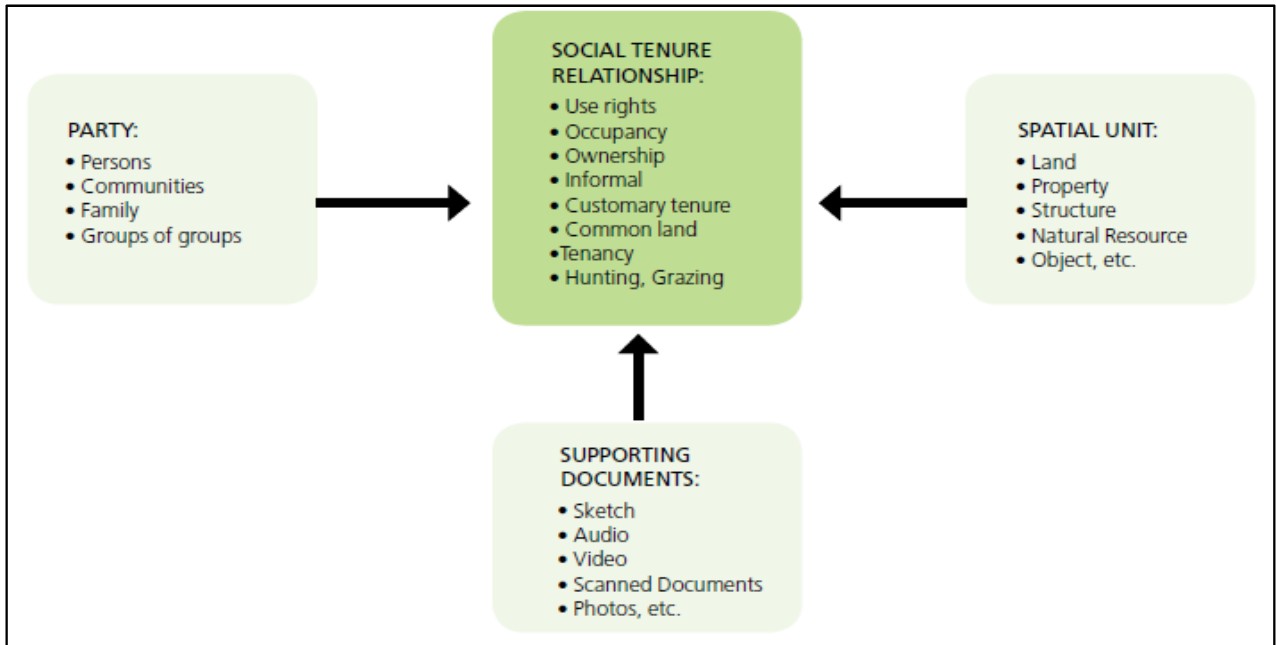

**Figure 1.** The conceptual model for the Social Tenure Domain Model (STDM).

The technical development of the STDM tool is being led by the Global Land Tool Network partners and UN-Habitat, and it is based on a popular and stable open-source QGIS and PostgreSQL/PostGIS database application. The authors in this paper have been involved in the design, testing, and implementation of the tool at the country level. Different formats of spatial and nonspatial data are supported using common exchange formats supported in many GIS and database applications. This also means that the STDM tool can be integrated with other applications and even formal land information systems through a common interface. The source code has been made available for wide access by developers to enhance the features and functionality. A stable version of the tool is maintained by UN-Habitat and the GLTN Secretariat, who also supports in the customization of the tool to suit different application contexts. The tool can also be used at various levels within an organization from a simple geo-information system to a complete land information system

with a centralized data repository utilizing client–server architecture to serve multiple users in different departments in an organization.

Moreover, the design of the STDM tool is oriented toward a pro-poor approach to support applications even with limited resources and technical skills. The data collection methodologies rely on the use of handheld Global Positioning System (GPS) receivers, satellite imagers, and smartphone applications, among others. Thus, the tool is promoting the use of low-cost solutions particularly in the development of the FFP LA spatial framework. Essentially, the tool aims to complement the existing land administration system in addressing the land registration gap [29]. The notable impact has improved the perception of tenure security and the easy access to information for planning and decision making [28].

Hence, the STDM contributes to the development of the FFP LA spatial framework through its flexible recording of data, participatory approaches, and application of low-cost technologies. This research sought to investigate, through empirical evidence, the impact of the STDM tool on improving land tenure security along the Continuum of Land Rights approach and its contributions to changes in policies and institutional arrangements. To test these impacts, this study hypothesized that the STDM tool facilitates tenure security improvement, ultimately impacting policy implementation and changes in institutional practices. The arguments in the discussion reflect on the case studies' observations and primary data collection to test the hypothesis.

## 2. Materials and Methods

The primary objectives of this study were to determine whether the application of the STDM tool facilitates the improvement of land tenure security along the Continuum of Land Rights Approach and whether it contributes to influencing positive changes in policy and institutional arrangements. To evaluate these objectives, the research work developed these specific research questions: Whether the STDM tool facilitates land tenure security improvement, and if so, how? Whether the STDM application contributes to positive changes in policy and institutional arrangements, and how? Other research questions were open-ended and Likert-scale questions were meant to further understand the emerging outcomes and to reinforce the above questions.

To respond to the research questions, a combination of qualitative and quantitative research methodology was used. Three case studies were considered where the STDM tool was implemented and/or is being implemented. The authors are familiar with them and have played a substantive role in the development and implementation of the STDM tool, which contributed to the design and conduct of the research and the subsequent analysis of the findings. A questionnaire was developed and administered through primary data collection to gather quantitative data. A purposeful sampling technique was considered to target respondents that played a specific role in the case studies, and the selection was guided by project partners in each country. The questionnaire was administered using an online survey tool called Survey Monkey (https://www.surveymonkey.com, accessed on 30 March 2021). Detailed information about the respondents' role is presented in Section 2.2. The observations and results from the primary data collection were then presented in the spreadsheet format, they were analyzed, and reflections were offered to particularly respond to the research questions. The study also used secondary data and the literature review technique to validate the primary data. The analysis of secondary data assessed the case study reports, papers, articles testimonies, and video documentation evidence to present key achievements and results.

### 2.1. Case Studies

2.1.1. Introduction

The case studies section presents the projects in brief, the application context, the processes involved, and the key achievements in each country. The use of the STDM tool was central to the project and the research attempts to explore its contribution and impacts in relation to the research questions. The three case studies were from Sub-Saharan Africa,

and the processes were driven by local communities and were supported by civil society organizations (CSOs) and local authorities in each country. The Figure 2 below shows maps of the three countries and the specific location where STDM was/ is being implemented.

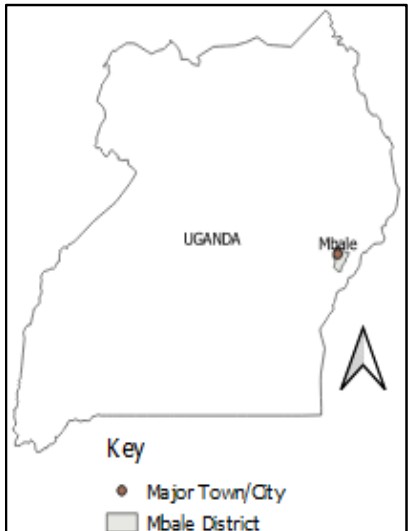 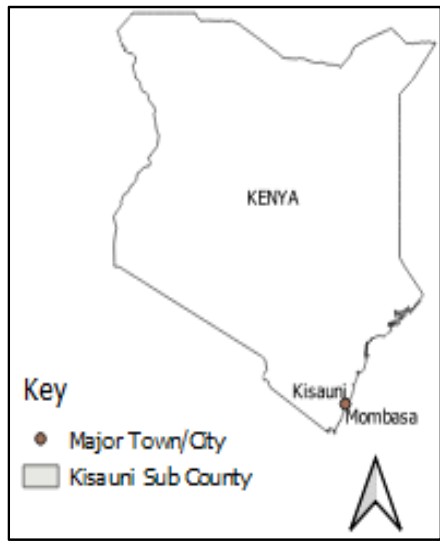 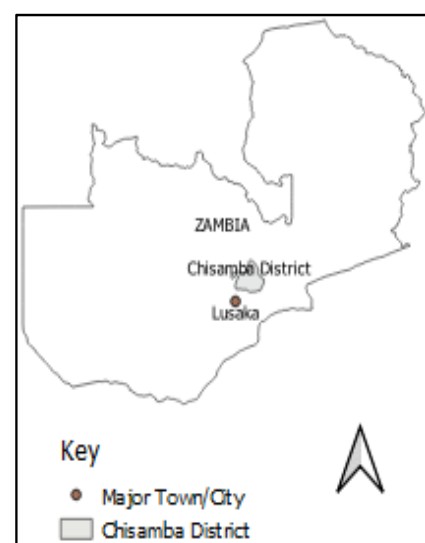

**Figure 2.** STDM case studies area in the three countries.

The initial activities involved planning and consultations with key implementing partners at the national and municipal levels to introduce the STDM tool and its objectives in the early stages of the project. A series of sensitization meetings with target communities and broader stakeholders, including joint need assessment missions, were conducted at the local level [23,31]. The concepts of the STDM tool were presented at the very start of projects, including the FFP LA principles. An enumeration questionnaire that captured access to land questions was developed and administered at the individual and household levels. The enumeration questionnaire also served as the tool for collecting specific information that was deemed relevant to the community, such as access to services, household information, and community priorities. The STDM tool was then customized to fit the local context. Communities were then trained to use satellite imagery and handheld GPS devices to demarcate settlement boundaries and land parcels (spatial units) in each context.

The data from the enumeration and participatory mapping processes were encoded and added into the STDM database. However, it should be noted that while this is a 'generic' procedure, the specific processes were always determined in consultations with partners and local communities. The data were handed over to local authorities for quality assurance and validation. Activities to support the basic maintenance of the database through the regular backup of data were performed by selected community members on a voluntary basis. UN-Habitat/GLTN also provided support through the provision of updated software whenever a new version of the STDM tool became available.

2.1.2. Uganda

Context and Interventions

Uganda is rapidly urbanizing. With an urban population growth rate of approximately 5.9% per annum as of 2019 and with close to 25% of the total population living in urban areas (approximately 11 million), urbanization in Uganda is set to further increase by over 30% by the year 2030 [32–34]. According to UN-Habitat, 48.3% of Uganda's population was living in informal settlements in 2018, a situation caused by the inadequate capacities of urban areas to absorb new populations who, in turn, found their own informal ways to establish themselves in the city [35]. The project 'Transforming Settlements of the Urban Poor in Uganda' (TSUPU) was launched by the Government of Uganda in 2011, through the Ministry of Lands, Housing and Urban Development (MLHUD) in partnership with

the Cities Alliance, aiming to address urbanization challenges in cities and towns. Among the projects provided for under the TSUPU project was the piloting of the Social Tenure Domain Model (STDM) tool and, later, its implementation and scaling up. Bufumbo and Mission cells in Namakwekwe Ward, covering approximately 28.26 hectares in the Northern Division within Mbale City (then a municipality), were selected as pilot zones based on the accessibility and proximity to Mbale town [23]. Local project partners were ACTogether Uganda and the grassroots organization, National Slum Dwellers Federation of Uganda (NSDFU), affiliated with Slum Dwellers International (SDI), an international CSO. The capacities of local youth and selected village representatives were enhanced on the use of the STDM tool and technical know-how on data analysis through a series of training sessions.

Key Achievements

The Mbale Municipal Council provided office space for managing the data in the STDM tool equipped with a computer, a printer, cameras, and furniture. The office space/resource center created a platform for regular interaction between the community and the municipal authorities. The community through the Mbale Slum Dwellers Federation was able to interrogate the data to identify gaps related to community development projects, issues of security, and access to services. The STDM generated data were used to lobby for services and other pressing needs in the community. For instance, in the subsequent phases of the TSUPU project, the local community applied for a grant to address the gaps identified from the STDM generated data. This led to the construction of eight (8) water points, five (5) public toilets, one (1) school toilet, and two (2) stone pitched drainage channels, as well as the widening of 1 road and the improvement of street lighting by Mbale Municipal Council [36]. A community center was also constructed in Mission cell with sanitation and office/meeting facilities, managed by the local community.

Further, citywide profiling, enumeration, and mapping were carried out covering fourteen (14) secondary towns of Uganda (including Mbale City), in which more than 181,000 households residing in approximately 120 informal settlements were enumerated and mapped, and their data were captured using the STDM [37]. The lessons and experiences from the TSUPU project have informed more recent projects in Uganda on the use of the STDM tool toward the legal recognition of land tenure security. The STDM tool and the participatory processes have been used to demarcate land at the household level. The data generated from the STDM tool is being used to issue Certificate of Occupancy, Certificate of Customary Ownership (CCO), and the application of freehold land titles, as provided for in the Land Act of 1998 and in the National Land Policy of 2013 [32,37]. The MLHUD in 2019 issued a decree on the use of the STDM tool to issue digital CCOs [38].

2.1.3. Kenya

Kenya's urban population as a proportion of the total population increased from 8.8% between 1960 and 1970 to 20.9% between 2000 and 2010, and it is projected to exceed by 36% between now and 2040 [39]. The growing trend of urban informality defines much of Kenya's urbanization patterns often recorded within informal settlements as the towns and cities grow [40].

The case study in Kenya looked at an informal settlement area called the Kwa Bulo settlement located in Kadzandani ward in Mombasa County. The settlement measures approximately 36 hectares and is divided into four clusters/villages: EPCO; Kashani; Timboni; and Msufini [38]. The first registered proprietor of the plot was Emmanuel Bulo, after which the land was transacted on and passed to numerous entities. In 2005, approximately seven hundred houses were demolished in an eviction orchestrated by the Kenya National Assurance Company [39]. In May 2006, residents of Kwa Bulo went to court seeking legal redress on grounds of adverse possession but had to wait until 2011 when the Kenya High Court of Mombasa issued a decision allowing the community to access the land.

Kwa Bulo community was then advised by the Mombasa County Government (MCG) to conduct an enumeration exercise that emulated a similar exercise conducted in Mnazi Moja, an informal settlement at the edges of Mombasa City, to identify the actual occupants of the land. Enumerations began in December 2014 led by community members in Kwa Bulo, Pamoja Trust, and Muungano Wa Wanavijiji, a social movement of 'slum' residents and urban poor people in Kenya and a partner to Pamoja Trust. Officials from the government collaborated with the community in data collection and data verification.

The community members performed data entry for all the administered questionnaires into the customized STDM tool. The data were later analyzed to produce a list of all beneficiaries in the settlement, including their relationship to the land and structures. The Kwa Bulo settlement land committee worked with representatives of Pamoja Trust and Muungano Wa Wanavijiji to verify the accuracy of the beneficiary list [31]. The County government also performed a similar validation exercise that involved officials from MCG, accompanied by community leaders of Kwa Bulo and Pamoja Trust representatives. After a successful validation exercise, the team concluded and accepted about 800 parcels for further processing.

Key Achievements

The MCG agreed to issue certificates of occupancy to the occupants that had complete and uncontested data. A total of 944 certificates of occupancy (397 to women, 542 to men, and five jointly/both spouses) were issued to residents of the Kwa Bulo settlement in 2017, benefitting approximately 3722 family members [36,41]. The issuance of certificates of occupancy improved the tenure perception of the community, gave them a sense of identity, and protected them from forced evictions [20,41]. Improved confidence in tenure security is being witnessed through housing improvements and better construction materials (ibid). A community data center was also established with furniture and computers that hosted the STDM database. A team of youths trained on the STDM tool were responsible for data management in the center. Local advocacy efforts by the community using the STDM generated data led to the installation of a clean water supply in the settlement [41]. Additionally, the council authorities are planning to build an access road, extend water lines, and install public toilets in the settlement. To date, the MCG is finalizing the process of transferring the land from the private landowner to the council and is having some discussions with the community on the next steps toward further strengthening their tenure security.

2.1.4. Zambia

Zambia has a dual land tenure system namely, leasehold, or statutory tenure and customary tenure type. At independence in 1964, customary land was said to account for approximately 94% of all land in Zambia [42–44]. As of 2005, customary land was estimated at 84% [45]. About 10% of customary land has been converted to private leaseholds since the enactment of the Land Act in 1995 [46,47]. Customary tenure in Zambia is an indigenous form of land ownership that is governed by unwritten traditional rules and administered by traditional leaders/Chiefs and Chieftainess. The active occupation or use of a piece of land is the main evidence of ownership or an existing interest in the land. Access to customary land is also contingent upon ethnic, kin, or community membership, controlled by the Chief or Chieftainess. This form of tenure is, however, not secure, because it is largely undocumented, making inhabitants of customary land susceptible to forced displacements, abuse from traditional authorities, threats from investors due to the increased demand of land, and frequent land disputes among villagers, head persons, and even between chiefs pertaining to land boundaries.

The case study in Zambia looks at Chamuka chiefdom, which consists of 208 villages in Chisamba District, Central Province, about 100 km from Lusaka city, the capital city of Zambia. Each village is headed by a village head person. The chiefdom covers a spatial extent of approximately 300,000 hectares of land. The discussion on the STDM implemen-

tation started in 2016 between the People's Process on Housing and Poverty in Zambia (PPHPZ), the Zambia Homeless People Federation (ZHPF), and the traditional leaders of Chamuka Chiefdom led by His Royal Highness Chief Chamuka VI, in collaboration with government authorities and other stakeholders with support from the UN-Habitat Country Office in Zambia.

Chief Chamuka had picked the interest of the STDM tool and its participatory processes while implementation was ongoing in the neighboring Mungule chiefdom in 2015, and they were very eager to welcome this process in his Chiefdom that brought together the traditional authorities in Chamuka, the local community, as well as government officials from Kabwe municipality in the implementation phase [48].

Key Achievements

The implementation of the STDM in Chamuka begun in 2016 and, so far, twenty-seven (27) villages have been enumerated and mapped. A total of 1794 households have acquired certificates of customary land occupancy representing a population of approximately 10,812 people, out of which 5789 are women beneficiaries. There are 765 joint ownership certificates, which means that the certificates are in the name of both spouses [41]. A community resource center has been established in Chamuka chiefdom, to be operated and managed by youths trained on the STDM tool. The center will also serve as an anchor to all STDM activities in the chiefdom. Over 85 para-surveyors were trained to ensure the sustainability of the STDM and related processes in other villages in the Chamuka chiefdom.

The community is using these certificates as proof of actual occupancy, giving them a voice in negotiating with investors who have expressed interest in their land. For example, between 2018 and 2019, seven (7) families in Bulemu village occupying a 103-hectare piece of land successfully negotiated with a private firm for a solar project in a meeting presided over by Chief Chamuka and the headmen of the Chiefdom. The signing of agreements between the investor and the families was completed in April 2019.

The implementation process has also improved the women's participation by giving them a conspicuous role in spearheading data collection and facilitating discussions surrounding land rights with the traditional authorities. His Royal Highness Chief Chamuka has also introduced a policy declaring that fifty (50%) per cent of the land at any given time be reserved for women in all the 208 villages [49]. Additionally, every village committee is required to have women representation to ensure women voices are heard. Some youth members from Chamuka chiefdom have also benefitted by being allotted a piece of land for an aquaculture project by the Ministry of Livestock and Fisheries [50].

Chief Chamuka, headmen, and villagers alike observe that boundary disputes that were rampant before the STDM enumeration and mapping processes are now a thing of the past [49]. A total of 426 land disputes (intra-family succession disputes, village boundary disputes, etc.) were recorded and successfully resolved. Thus, the STDM tool and related processes have become instrumental in resolving long-standing disputes in the Chiefdom. Moreover, the experience from Chamuka chiefdom has greatly contributed to the further development of the draft National Land Policy. Specifically, the recommendations from Chamuka provided evidence on the practicality and versatility of fit-for-purpose land tools such as the STDM to enhance tenure security on customary land for all [25]. The final draft of the national land policy has prominently provided for the recognition of customary land rights through the Chiefs and the promotion of fit-for-purpose land administration approaches. The National Land Policy was approved and adopted by the Government of Zambia in May 2021.

*2.2. Primary Data Collection*

The primary data collection survey was conducted in October and November 2020 and targeted government authorities, CSOs, traditional authorities, members of enumeration, mapping teams, community leaders, and members within the case study areas. The

intention of the survey was to gather views and opinions of the various beneficiaries and stakeholders around the use and application of the STDM tool, particularly in relation to its contribution to improving tenure security and any indications that demonstrate its influence on policy and institutional arrangements. Aside from online surveys, and due to limited internet access and the need for translation for some respondents, some questionnaires were filled manually by some respondents. The partner CSOs supported in reaching out to the target respondents and assisted in the administration of the questionnaires, particularly in translation into local languages.

From Table 1, a total of 82 participants responded to the survey in the three case studies and comprised 50% women and 50% men; and about 80% of the respondents were community stakeholders. All respondents participated in STDM interventions for a period ranging from two to eight years, and 80% of them are still involved in the ongoing work at the time of the survey.

**Table 1.** Respondents for the primary data collection.

| Respondents | Country | | | Percentage |
|---|---|---|---|---|
| | **Kenya** | **Uganda** | **Zambia** | |
| Local government official/staff | 0 | 1 | 2 | 4% |
| Traditional authority | 0 | 0 | 4 | 5% |
| Community leader | 8 | 4 | 5 | 21% |
| Community member | 9 | 8 | 8 | 30% |
| Member of enumeration/mapping team | 3 | 10 | 7 | 24% |
| Member of CSOs | 4 | 6 | 3 | 16% |
| Total | 24 | 29 | 29 | 100% |

*2.3. Challenges and Limitations of the Primary Data Collection*

The observation from the responses was that they were quite consistent across the three countries. As the questionnaire was administered online, we could not interact with the respondents to follow-up on the research questions that would have improved the analysis of the quantitative data.

**3. Results**

The analysis of the survey entailed compiling the results using the survey monkey tool and then exporting the data into a spreadsheet format. The data were organized and categorized into a tabular format and were grouped according to the research questions. Conversion of the data into a spreadsheet allowed for more discrete analysis.

*3.1. The STDM Tool Facilitates Land Tenure Security Improvement*

We asked the respondents to state whether the STDM tool facilitates tenure security improvement. According to the results, 100% of the respondents from Zambia and Uganda agreed with the statement that the STDM tool facilitates tenure security improvement, while for Kenyan respondents, the endorsement of this statement was at 83%. From Table 2 below, 95% of all respondents from the three countries acknowledged the statement, with 42% of respondents strongly in agreement. Moreover, we analyzed the responses based on gender and found that over 97% of the men and 93% of women agreed that tenure security has improved in the target community, as a result of the STDM-related interventions.

**Table 2.** Responses on whether the STDM facilitates tenure security.

| Description | Number of Responses | | | Percentage |
|---|---|---|---|---|
| | **Kenya** | **Uganda** | **Zambia** | |
| Strongly agree | 10 | 12 | 12 | 42% |
| Agree | 10 | 16 | 17 | 53% |
| Neither agree nor disagree | 3 | 0 | 0 | 4% |
| Disagree | 1 | 0 | 0 | 1% |
| Strongly disagree | 0 | 0 | 0 | 0% |
| Total | 24 | 28 | 29 | 100% |

We also asked the respondents to indicate what could be the reasons for such perceptions of improved tenure security. There are at least three key common responses to this question across the countries: reduced land disputes and conflicts (81%), increase in the overall confidence and perception of tenure security (78%), and receipt of the certificate of occupancy (74%). For Zambian respondents, their top two responses were: increase in the overall confidence and perception of tenure security (96%), and receipt of tenure documents (82%). Meanwhile, in Kenya, the respondents chose reduced land disputes and conflicts (68%), and available/access to settlement and household data and maps (63%) as their main indications of the increased perception of tenure security. For Ugandan respondents, reduced land disputes and conflicts (96%) and a decreased risk of forced evictions in the community (88%) were the key elements of having an improved perception of tenure security.

*3.2. Use of the STDM Tool Contributes to Positive Changes in Policy and Institutional Arrangements*

We then sought to determine the respondents' view about the STDM tool being able to contribute to influencing changes in related land policies and institutional arrangements. The result indicated that 100% of all the respondents from Zambia and Uganda agreed with the statement that the STDM tool was able to influence related land policies and institutional arrangements. Interestingly, Kenyan responses only indicated an agreement of 74% with the question. Table 3 below represents the perception from the three countries. Overall, 93% of the total respondents agreed with the statement, of which 38% strongly agreed.

**Table 3.** Perceptions on the STDM ability to influence changes in policy and institutional arrangements.

| Description | Number of Responses | | | Percentage |
|---|---|---|---|---|
| | **Kenya** | **Uganda** | **Zambia** | |
| Strongly agree | 7 | 9 | 14 | 38% |
| Agree | 10 | 19 | 15 | 55% |
| Neither agree nor disagree | 5 | 0 | 0 | 6% |
| Disagree | 1 | 0 | 0 | 1% |
| Total | 23 | 28 | 29 | 100% |

Tweaking the data to assess the responses by gender, the data showed that 57% of the women expressed confidence compared to 52% of men who also agreed that the STDM tool had an impact on influencing changes in policy and institutional arrangements.

Moreover, we tried to find out how the application of the STDM tool had influenced policies and institutional arrangements. The common responses (above 50%) included the following: marked improvement in the relationship between the community and authorities, increase in the demand for similar interventions in other areas, recommendations for the wider application of the STDM through potential changes and institutional arrangements and stakeholders, including traditional and local authorities being supportive of the STDM tool implementation and its replication/scaling-up. While conforming to the above

trend of the responses, each country, however, had a different emphasis. For example, we found out that 66% of Kenyan respondents felt that having the different stakeholders being supportive of the STDM tool implementation and its replication is the most important manifestation that it is influencing policies and institutional arrangements. More than 80% of the respondents from Uganda agreed with the views from Kenya, but also included two other relevant factors: the improved relationship between the community and local authorities and the better engagement and support of local authorities in improving the settlement. In Zambia, more than 65% of the responses pointed out that there is a strong recommendation for wider application of the STDM tool application through changes in policies and institutional arrangements, as well as increased demands for similar interventions in other areas.

### 3.3. Other Results

We asked the respondents to give their opinion on whether the community is better off after the STDM-based interventions. The results in Table 4 below show that 95% of the respondents in the three countries agreed that the community is better off after the STDM-based interventions, with 41% of the respondents were strongly in agreement with the statement.

**Table 4.** Perceptions that the target community is better off after STDM interventions.

| Description | Number of Responses | | | Percentage |
|---|---|---|---|---|
| | **Kenya** | **Uganda** | **Zambia** | |
| Strongly agree | 10 | 12 | 12 | 41% |
| Agree | 10 | 16 | 17 | 54% |
| Neither agree nor disagree | 3 | 0 | 0 | 4% |
| Disagree | 1 | 0 | 0 | 1% |
| Total | 24 | 28 | 29 | 100% |

Perception questions were asked and one of them referred to whether the respondents would recommend replicating or scaling up the STDM implementation in other areas. This question received a positive response of 95%, with 54% of respondents strongly in agreement. Another related question was asked on whether the respondents would consider the STDM tool as an innovative and useful tool in addressing land tenure and land governance. About 91% of the respondents agreed with the proposition, with 46% strongly in agreement. Finally, the respondents were asked why they thought the STDM was a useful and innovative tool. The respondents highlighted at least three justifications for their answers, namely: low-cost and affordability (73%); inclusive, gender-responsive, and fit-for-purpose (73%); and the process is empowering (70%).

## 4. Discussion

It can be clearly shown that the STDM tool contributed to facilitating improved perceptions of tenure security along the continuum, and to a significant extent, to influencing policy and institutional arrangements. It is important to note, however, that the implementation of the STDM tool was complemented by active community engagement and the use of other tools such as participatory enumeration, and the gender evaluation tool also developed by GLTN.

### 4.1. STDM Tool as Facilitating Tenure Security

To contextualize the discussion on tenure security in this research, we considered the continuum of land rights approach in interpreting the findings. The results indicated that through the STDM application, tenure security of the community has improved. This was mainly attributable to: the increased confidence through participation; the availability and accessibility of the data; the reduction in land disputes and conflicts; the issuance of tenure

documents; the decreased risk of arbitrary forced evictions; the improved recognition and relationship with authorities. While these indicators are evidently present in all three countries, each case study has different hierarchical views and emphases, as indicated above. Potentially, the indicators of the improved perception of tenure security will evolve as new events, situations, or variables occur along the continuum of land rights approach.

Likewise, there was recognition that land tenure documents were issued at reduced costs compared to formal tenure documents (such as freehold titles), which are normally costly for the beneficiaries and for the government authorities. The issuance of the occupancy certificate provided a sense of legitimacy to the *de facto rights* of communities in that the rights are recognized by authorities and the certificates can be used to settle claims. It is important to note that women received the same tenure documents in a similar manner as men. We also noted the high perception of tenure security among women, and we attributed this to their active participation in the enumeration and mapping processes, including in the data validation. Hence, the process had large implications in their identity as bona fide occupants or claimants of the land.

There was also strong evidence that land conflicts or disputes were reduced. This could be attributed to the availability and accessibility of data at the settlement level and the strong community engagement and participation in the whole process. Furthermore, we observed that anyone in the community could check and verify, and that the preliminary data were validated by the community, which facilitated data acceptance, data accuracy, and the collective resolution of disputes. In Kenya, validation was done at three levels before the data could be approved for the issuance of occupancy certificates. In all three case studies, the disputes were resolved in the presence of traditional and local authorities, elders, and neighbors as witnesses, and all pertinent documentation was attached to the information and were recorded in the STDM database.

### *4.2. STDM Application as Contributing to Influencing Change in Land-Policies and Institutional Arrangements*

#### 4.2.1. Influence on Policy Development and Implementation

In Zambia, the case study experiences impacted the policy development process to a level that the FFP LA approach was included in the current draft national land policy to speed up customary land registration, among others. From the case study, we learnt that land policy consultations with traditional authorities had increased, and this was also shown in the survey finding through increased demand for similar interventions in other regions.

It is notable that the experiences of the STDM tool in the case studies resulted in its increased uptake in other contexts and the scaling up of the interventions in other areas. For instance, in Uganda, the roll-out of the tool from Mbale City to other regions, and later on, to customary land registration processes, has resulted in its adoption by the Government of Uganda in digitalizing the processes and the issuance of Certificates of Customary Ownerships (CCOs) and related processes.

In Kenya, after the issuance of Certificates of Occupancy to the residents, perceptions of tenure security increased, and even the housing conditions that resulted in the County Government increased investments in settlement upgrading plans and incremental developments, also building from the stronger relationship that was established with the local community. The slum upgrading process is guided by the national slum prevention and upgrading policy of Kenya. The policy supports community organization toward a common goal through the establishment of an information system that records community resources and rightful beneficiaries. This step has already been achieved. As a result, the MCG is now negotiating with the private landowner so that they can officially procure the Kwa Bulo land for future planning, the upgrading of the settlement, and the further strengthening of tenure security of the residents.

### 4.2.2. Changes in the Institutional Arrangements

The outcome of the country case studies clearly demonstrated positive changes in the institutional arrangements particularly between the communities and local and traditional authorities. The authorities' acceptance to offer office space for storing and managing community data demonstrates improved trust, confidence, and relationships among the local stakeholders. In addition, local and traditional authorities and community members have equal access to the information. Thus, access to data facilitated an open-door policy for interactions, dialogues, and consultations in the planning and decision-making processes.

Similarly, the use of the STDM tool improved accessibility to the data and gave the community the opportunity to negotiate on matters related to service delivery, local investment in infrastructure, settlement planning, and development projects with authorities. These opportunities further contributed to community empowerment. At the time of this research, the negotiated projects have already been completed or are ongoing in the three case studies. With the strong partnership with local and traditional authorities, the communities were able to advocate and articulate their issues and concerns through them, which translated to some concrete actions in influencing changes in policy and regulatory frameworks.

### 4.3. STDM Technology and Related Processes Supporting Community Empowerment

The use of the STDM tool has benefited the community in building collective knowledge of their settlement. It has also enabled them to organize themselves better in addressing community challenges such as identifying priorities, resolving land disputes, and strengthening their negotiation with authorities and with external stakeholders (e.g., investors).

It is also important to note that the participatory approach employed in the STDM tool implementation contributed to the 'ownership' of the projects through active involvement in the enumeration activities, community mobilization, and participatory mapping, thus contributing to the overall community empowerment. In addition, community members, particularly women, have a deeper sense of ownership of the processes, and it has raised their position to be part and parcel of decision making in the community. The process of localizing the related methodologies benefitted the community members, particularly the youth, from training and access to technology and information, which allowed them to manipulate and analyze the data in the STDM database.

In Zambia, the transformative change at the community level is clearly demonstrated, where several households were able to properly negotiate with investors on the planned development projects. Previously, such negotiations were concluded only by the traditional authorities without the consent and knowledge of community members. The use of the STDM tool has also scaled up in Zambia to urban areas, including increasing demands for capacity development interventions. Similarly, in the Kwa Bulo settlement, they were able to directly negotiate with the County Government to act on their behalf, issue occupancy certificates, and increase investments for settlement improvement.

### 5. Conclusions

The study investigated whether the STDM tool facilitated the improvement of land tenure security and whether it has contributed to influencing positive changes in policy and institutional arrangements in case study areas. This research has confirmed positive responses to these questions. There was remarkable improvement in the perception of tenure security in the three case studies linked to the STDM tool implementation along with the range of recommendations regarding the various actions, steps, and decisions on land that can be interpreted as having meaning to land tenure security. It is also clear that the use of the STDM tool is more effective and sustainable because it integrates participatory processes along with other tools in the implementation process. This approach, together with increased community empowerment and access to information for planning and decision-making, contributed positively to influencing positive changes in policy imple-

mentation and institutional arrangements. While the changes are not happening quickly, the indications are clearly demonstrating positive outcomes occurring in an incremental but steady manner that is sustainable at the local level.

The research has shown that technology such as the STDM tool can be effective toward the development of the FFP LA spatial framework and in facilitating the land tenure security of poor communities. The research also concluded that focusing on one pillar of FFP LA can influence positive changes in other frameworks of FFP LA through the use and application of technology such as the STDM tool. The study also contributes to the knowledge that the development of the FFP LA spatial framework can be achieved through the combined elements of adopting fit-for-purpose technology such as the STDM tool, awareness and capacity building, participation, engagement of local communities, and political support from local and traditional authorities. Therefore, there is a potential for more research to be carried out to demonstrate how one of the frameworks of FFP LA can catalyze changes in other frameworks. More related research on the topic is needed to further understand the dynamics of the various FFP LA frameworks and to strengthen the findings of this study.

**Author Contributions:** Conceptualization, D.A., S.N., H.N. and J.G.; methodology, D.A. and S.N.; software, S.N.; validation, D.A., S.N., H.N. and J.G; formal analysis, D.A. and S.N.; resources, D.A.; data curation, D.A. and S.N., writing—original draft preparation, S.N., D.A., H.N. and J.G.; writing—review and editing, S.N., D.A., H.N. and J.G.; visualization, S.N. and D.A.; project administration, S.N. and D.A.; funding acquisition, D.A. All authors have read and agreed to the published version of the manuscript.

**Funding:** Funding support for the research work including data collection is provided by UN-Habitat and Global Land Tool Network (GLTN) programme. The funding of the publication costs for this article has kindly been provided by the School of Land Administration Studies, Faculty ITC, from the University of Twente in combination with Kadaster International, the Netherlands.

**Institutional Review Board Statement:** The study was conducted according to the guidelines of the Declaration of Helsinki, and approved by the Institutional Review Board (or Ethics Committee) of UN-Habitat and the GLTN.

**Informed Consent Statement:** Informed consent was obtained from all subjects involved in the study.

**Data Availability Statement:** All the case study materials are referenced in the manuscript.

**Acknowledgments:** The study is based on the findings of UN-Habitat and the GLTN projects in the case studies in all three countries. The authors would like to acknowledge the contribution of the partner organizations in each country, Pamoja Trust in Kenya, ACTogether Uganda in Uganda, and People's Process on Housing and Property in Zambia, for their support in data collection.

**Conflicts of Interest:** The authors declare no conflict of interest.

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
