# Peer review of "Transforming Land Administration Practices through the Application of Fit-For-Purpose Technologies: Country Case Studies in Africa"

_land, doi:10.3390/land10050538_

Round 1

Reviewer 1 Report

The article studies the benefits of a tool to register land property, called “Social Tenure Domain Model (STDM)”. The authors follow the implementation of the tool in three different African countries and underline the potential of the tool to resolve conflicts over land attribution. The tool facilitate the registration of land ownership as well as land tenure security, and respond to a number of issues, such as competition for land or land grabbing, which are particularly relevant in the cases studied. Although there are many interesting ideas in the paper and we see how much the authors want to convince of the potential of the STDM tool, a number of issues remain and should be addressed before publication.

First, the theoretical framework of the paper could be clarified. Some of concepts used are not well presented, such as the “Fit-For-Purpose Land Administration” or “the continuum of land right approach”. The authors should not assume that the readers know what these very specific concepts are and which discussions they refer to, but instead introduce and unpack them.

Second, the methodology of analysis of the paper should be more extensively presented. The authors need to say how the data was analysed. What is the method used here? For the questionnaire presented in the second part of the paper, they need to better explain what were the questions and the choices available, and again how the data was analysed. This should be presented in a methodological section at the beginning of the paper (the 2.2 primary data collection comes too late and should be further developed). The authors could also reflect on the potential and limits of this methodological framing.

Third, some results should be supported with more evidence. I felt that the ‘emerging outcomes’ attributed by the authors to the STDM tool in the two first case studies were less justified than in the last case. We need to know exactly which changes can be attributed to the STDM tool and why? For instance, it is not enough to say that relations were improved, but how did this happen with the STDM tool? The authors need to convince us that this happened because of their intervention and not because of other factors.

Taking the example of the first case,

“The Mbale Municipal Council provided office space for the learning centre, equipped with a computer, a printer, cameras, and furniture for communities to use in managing the data regularly. As a result, there was an improved relationship between the community and the municipal authorities with an open-door policy regarding project interventions among other consultations. The community through the Mbale Slum Dwellers Federation was able to use the data reports to identify gaps related to community development projects, issues of security and access to services. The data was also used to lobby for services and other pressing needs in the community. For instance, in the subsequent related phases of the project, Mbale Slum Dwellers Federation received grants to implement community projects aimed at improving the living conditions in the informal settlements which led to the construction of eight (8) water point projects, five (5) public toilets, one (1) school toilet; two (2) stone pitched drainage channels, widening of 1 road, and improved street lighting. A community centre was also constructed in Mission cell with sanitation and office/ meeting facilities and is managed by the local community”.

Here, they need to better justify the causality of each proposition and the relevance of the examples mentioned to support their arguments. For now, I am not convinced that the changes resulted because they used the STDM tool. I am much more convinced by the following example of this section on the use of tool “for the legal recognition of their lands” because I can understand here how this was made possible by using the STDM tool. Again, the arguments are well justified in the Zambia case and we expect to see similar consistency in the other sections.

Fourth, I have doubts about the added value of the results of the questionnaire as they are presented now. It seems that they mostly confirm that people are happy with the STDM, but they do not say what really happened. One possibility to improve this section would be to further reflect on the differences between the cases and social, historical and administrative particularities, to try to tease out why the results presented are different in the three cases. Also I am not sure that the statistical significance of one result is relevant, because this is the only passage where this kind of analysis is used and it does not really contribute to the general argument of the paper.

The discussion section is quite good and manages to answer to the questions of the paper by summarizing the results.

Some additional remarks:

“Does STDM application contribute to positive changes in policy and institutional arrangements? How does the STDM application, through the development of the FFP LA spatial framework, contributes to positive changes in policy and institutional arrangements?”

The two questions are a bit similar. I would advise to remove the second question, especially since the FPP LA has not been properly introduced for readers not familiar with it.

“The online survey was conducted using the Survey Monkey tool targeting responses from carefully selected informants.”

if they say that they need to say how the informants were selected. In addition, we do not know what is the survey monkey tool, and this needs to be presented.

Author Response

Many thanks for taking time to review this manuscript. Your substantial review and comments have contributed to further improvement of the paper. It was our pleasure to read them and provide feedback.
We have done our best to address all your comments and recommendations, and where we may have not done so, bear with us and we take note of that to be considered in future work because of limitations expressed in the paper. However, we are happy to hear from you again in further revisions of this paper where possible.

Reviewer 2 Report

General comment.

The manuscript presents the evidence well and it is well structured. The conceptual framework and referencing using the existing journal papers needs to be strengthened. The authors need to provide an overview of some of the international literature that deals with LAS and security of tenure to show why STDM is some kind of proxy to that system and therefore FFP. Also there are many journal articles on land about what provides security of tenure; gender issues (e.g. Meinzen-Dick); about participatory processes; and the link between dispute resolution decrease with respect to increases in security of tenure (e.g. Deininger in Uganda). The number of references need to be expanded to strengthen its scientific argument.

Specific comments

  1. Line 224-225 - it is unclear why there is a one line reference to customary
  2. More published journal references needed describing Zambian customary tenure
  3. Line 466 - define continuum of land rights and link the concept to the statements that follow
  4. Line 485 Need to make the Payne argument clear
  5. Line 514 - how do the authors know that the County government is increasing its investments in informal settlements - or is this an assertion
  6. References 5 and 16 are the same

Author Response

(The authors gave the same response as above.)

Reviewer 3 Report

This paper analyses the impact of the Social Tenure Domain Model on perceived tenure security and on the developments of Land Policies. It is a well documented, interesting to read and a very accessible paper that needs to be published.

Further improvement of the paper is possible. The issues listed here below may be supportive to this.

Abstract: “Fit-For-Purpose spatial framework needs to be adapted…” >>> “The Fit-For-Purpose spatial framework needs to be adapted…”. See also line 39: “United Nations General Assembly…” >> should be “The United Nations General Assembly….”. Please check the whole paper on this. Also: “STDM” should be “the STDM”, “GLTN” is “the GLTN”, etc.  Pay also attention to single/plural: “various level” in line 94 should be “various levels”. Please check the whole paper on this - there are many examples like this. This is not nice in a scientific paper...

Line 6: “particularly in is” something is missing?

Line 41: “land tenure gap,…” >>> this could be better explained in relation to land registration

Line 66: “technology like STDM,..”” >>> STDM is a technology? This word is used in many places. Is it a tool?

Line 119 -126:  should there be attention to gender here? Because that aspect is in the questionnaires

Line 124: “Other related research questions…” >>> why are those other questions not included? Please include or leave this sentence out

Line 130 -134: “The authors’ familiarity and substantive role in the development and implementation of the STDM tool, including the application of the tool in the three case studies supported the design and implementation of the research and the subsequent analysis of the findings.” >>> This is on more method: “Action Research”....

Line 185: “other secondary cities…”>>> why “other”? So far no secondary city is mentioned

Line 235: “acres of land”>>> what is this in sq meter? Mention area of the site in Uganda too. Later the unit hectares is used

Line 241: “response from the Kenya High Court” >>> explain please

Line 258: “parcels” >>> could be “spatial units”? Or was there a formal right established? That could be explained then

Line 261: “to those properties” >>> you mean “to this spatial units”??

Line 262 – 264: 944 certificates to 3,722 households? Please explain. Same remark applies to the Zambia case Line 301 – 312

Line 338 – Line 345: impact on policies. Similar experience in the other cases?

Line 348: “primary data collection survey targeted” >>> when was this done?

Line 361: “are still active” >>> when?

Line 376: “on another related questions,..” >>> which question?

Line 380: table numbering is incorrect. Also for other tables that follow. Please place the countries in the same order as in table 1. This makes comparison more easy

Line 382: “follow-up question,…” >>> were the possible answers to this question predefined or open? Then the results could be in a a table?

Line 382 – 393: “reduced conflict” is mentioned several times. Very nice! But what does that mean?  Substantial reduction? No conflicts left? More details are in lines 487 -495. But we don’t know this already here...

Line 400: “…that STDM work was able to influence related land policies…” >>> this means that this influence really exists – not only in Zambia as we could see – but also in the other countries. Please explain in more detail

Line 408 – 409:  women/men distinction >>> could this distinction provide relevant extra information for the other questions?

Line 412: “(p<.002).” >>> explain please

Line 414: “…common responses ….” : predefined answers? Then the results can be in a table?

Line 438 – 448:  Please present the results in table?

Line 456 – 463:  not needed, can be shorter – and include link to research questions?

Line 504 – 505: “result of the survey also showed” – where is this presented?

Line 506 – 520:  what is mentioned here could have been indicated under the cases studies for Uganda and Kenya. Are there references for the statements in lines 501 – 521?

Line 529: “normal circumstances,” >>> explain please. What are "normal circumstances?

Line 535 – 536:  “to date” >>> when?

Line 471: “indicators” >>> this is new. Explain please? Which type of indicators (line 473? “indicators on the improved perception of tenure security?”). Is this included in the methodology?

Line 476: “minimal cost” >>> could you extend on this?

Line 479: “a sense of legitimacy” >>> ?? Please explain what this means?

Line 483: “had huge implication in their identity as bona fide occupants” >>> How do you know this?

Line 580 -581: “pillar” >>> you mean “framework”?  Or “principle”?

Line 582: “research is trying to suggest” >>> Rephrase please…

Line 588: “there is a potential for more research” >>> is this a recommendation?/

In general the word “sustainable” could be in the conclusions?? Because of the institutional impact?

Author Response

(The authors gave the same response as above.)

Reviewer 4 Report

This is a good paper with some excellent results from the analysis of the case studies. However, it requires further work to place the case studies and results within the wider FFP LA context.

General comments

  1. This is a great article about reviewing the impact of using the STDM tool in four case studies.
  2. The use of the STDM tool is perceived to be a pro-poor land recordation system approach that can support small settlements to record their land rights and increase their security of tenure. However, the FFP LA concept is focused on delivering LA solutions at scale to deliver national solutions. You need to clearly describe the role of the STDM tool within this wider concept of FFP LA and explain role of the STDM tool in delivering the national solution.
  3. If you consider that there are other tools comparable to the STDM tool available then these should be mentioned.
  4. The term ‘STDM’ is used in many contexts within the article: a data model, a tool for recording land rights, an application for managing land rights. The language around the use of these terms is loose and needs to be tightened.
  5. Within the ‘FFP LA Guiding Principles for Country Implementation’ it was recommended that the perimeters of customary tenure land should be recorded and safeguarded within the formal LA system and any future external land transactions agreed with local / national government. Was this approach considered?
  6. The FFP LA concept includes three frameworks. Your focus is on the spatial framework and you mention the institutional framework (that includes policy), but you do not mention the legal framework. If there were legal aspects to your study then these should be included.
  7. One key objective of FFP LA is to ensure that it is inclusive of all tenure types and that they can be recognised within the formal LA. Did you consider engaging with the National Land Registration and Cadastral Agency to collaborate over the type on land registration and cadastral data being collected with the STDM tool to ensure that recorded land rights could be more easily formalised with the National Land Registration and Cadastral Agency?
  8. A key part of the case studies was to determine if there were influences on the land policy and institutional arrangements. This influence would be biased if there were representatives of these institutions involved in the case studies. This should be explained and at what level, i.e. national, regional, municipal.
  9. Little is explained in the article about the on-going maintenance of the land rights data that has been collected and the sustainability of the solutions implemented, especially from a financial perspective. This should be explained.
  10. How was the land rights data managed and quality assured to ensure that the sensitive data was secured from a data management perspective and illegal changes to the data avoided?
  11. The adjudication process has only been briefly mentioned in lines 493-495. More detail should be provided, and any lessons learned.
  12. Lines 582-584: ‘This research is trying to suggest that it is not necessary to have all the FFP LA frameworks implemented simultaneously to effect change’. In the context of the local land recordation examples provided then this statement may be true. However, in the context of national level implementation then this statement is incorrect. All three frameworks need to be in place to achieve the objectives of FFP LA, especially at scale and in short timeframes.

Specific comments:

  1. Lines 94- 95: It is stated that ‘The tool can also be used at various level within an organization from a simple geo-information system to a complete land information system with a centralized data repository utilizing client-server architecture to serve multiple users in different departments within an organization.’ Is this theoretical or are there examples of this level of implementation?
  2. 1.1 Introduction: Only the STDM tool is mentioned here. However, under 4. Discussion you mention a number of other tools used in the study. The full set of tools should be described in the introduction and their uses and relationships explained.
  3. Line 386: ‘tenure documents’ are mentioned. You need to explain who issued these documents and if they are formal.
  4. The table numbering is incorrect, and the references need also to be corrected.
  5. Line 310: Only 27 out of an expected 208 villages were enumerated and mapped over a 3-year period. The reasons behind the slow progress should be explained and how the timeframes could be improved.
  6. Lines 275-276: ‘having some discussions with the community on the next steps towards further strengthening their tenure security’. What are these next steps?
  7. Lines 442-443: ‘consider the STDM tool as an innovative and useful tool in addressing land tenure and land governance’. How would the participants know that it is innovative if they have nothing to compare it against?
  8. Lines 476-478: ‘land tenure documents were issued at minimal cost compared to formal tenure documents (like freehold titles) which are normally costly for the beneficiaries and for the government authorities’. It would be good to have explicit costs for each case study.

Author Response

(The authors gave the same response as above.)

Round 2

Reviewer 1 Report

I thank the authors who took into account my remarks concerning: 1) a better presentation of the theoretical framing and notions used for non-specialists; 2) a better presentation of the methods (this could be more justified, but probably fits to the readership of this journal); 3) stronger evidence for the results presented.

Author Response

Many thank you for your continued follow up on the responses. We are excited you are satisfied with our responses.

Please note that we appreciate your understanding in the limited space to majorly alter our research due to heavy requirements in the formatting and limited size of the paper. We are however, taking note of all these recommendations and we welcome additional thought to improve this in future.

Reviewer 2 Report

Line 50. The frame supplied by the Payne reference regarding the actions, steps and decisions on land needs to be further explained. This frame should be tied better to the key questions raised under Materials and Methods (line129-131). These questions are addressed in the conclusions ( line 558-584). Consider also addressing the Payne frame in the conclusions.

Author Response

This is well noted and we appreciate your thought and suggestions to make the ideas flow logically. Please, find our inputs on the paper and we totally appreciate your understanding to our minimal inputs and additions.

On page 60-70, we have reviewed the concept of Payne.

468-474. We emphasize the focus on the continuum of land rights approach (Discussion section)

566-570. We emphasize the focus on the continuum of land rights approach (conclusions section).

Kindly, find our inputs on the revised manuscript.